# *MCC* Gene Silencing Is a CpG Island Methylator Phenotype-Associated Factor That Predisposes Colon Cancer Cells to Irinotecan and Olaparib

**DOI:** 10.3390/cancers14122859

**Published:** 2022-06-09

**Authors:** Zeenat Jahan, Fahad A. Benthani, Nicola Currey, Hannah W. Parker, Jane E. Dahlstrom, C. Elizabeth Caldon, Maija R. J. Kohonen-Corish

**Affiliations:** 1Woolcock Institute of Medical Research, 431 Glebe Point Road, Glebe, Sydney, NSW 2037, Australia; zeenat.jahan@sydney.edu.au (Z.J.); hannah.w.parker@student.uts.edu.au (H.W.P.); 2Garvan Institute of Medical Research, Sydney, NSW 2010, Australia; f.benthani@gmail.com (F.A.B.); nicolacurrey@msn.com (N.C.); l.caldon@garvan.org.au (C.E.C.); 3St. Vincent’s Clinical School, UNSW Sydney, Sydney, NSW 2010, Australia; 4Faculty of Science, University of Technology Sydney, Sydney, NSW 2007, Australia; 5ACT Pathology, The Canberra Hospital and Australian National University Medical School, Canberra, ACT 2605, Australia; jane.e.dahlstrom@act.gov.au; 6Microbiome Research Centre, School of Clinical Medicine, UNSW Sydney, Sydney, NSW 2217, Australia; 7School of Medicine, Western Sydney University, Sydney, NSW 2560, Australia

**Keywords:** colorectal cancer, precision medicine, epigenetic biomarker, mutated in colorectal cancer (MCC), CIMP

## Abstract

**Simple Summary:**

DNA hypermethylation of specific regulatory regions causes gene silencing that is an important cancer-promoting mechanism. A subset of colorectal cancers display concordant hypermethylation and silencing of multiple genes, and this appears to change the way in which tumors respond to some cancer therapies. The aim of this study was to evaluate how the presence of the *MCC* gene silencing relates to the highly methylated subset of colorectal cancers and how it may affect therapy responsiveness. We found that strong *MCC* silencing is found throughout the hypermethylated subset, but *MCC* expression is also lost or reduced in some other tumors which show hypomethylated regions of the gene. In cell culture experiments, the deletion of *MCC* increased the responsiveness of cancer cells to the chemotherapy drug irinotecan (SN38), and this was further augmented by a targeted cancer drug, the PARP-inhibitor Olaparib.

**Abstract:**

Chemotherapy is a mainstay of colorectal cancer treatment, and often involves a combination drug regime. CpG island methylator phenotype (CIMP)-positive tumors are potentially more responsive to the topoisomerase-inhibitor irinotecan. The mechanistic basis of the increased sensitivity of CIMP cancers to irinotecan is poorly understood. Mutated in Colorectal Cancer (*MCC*) is emerging as a multifunctional tumor suppressor gene in colorectal and liver cancers, and has been implicated in drug responsiveness. Here, we found that CIMP tumors undergo *MCC* loss almost exclusively via promoter hypermethylation rather than copy number variation or mutations. A subset of cancers display hypomethylation which is also associated with low *MCC* expression, particularly in rectal cancer, where CIMP is rare. *MCC* knockdown or deletion was found to sensitize cells to SN38 (the active metabolite of irinotecan) or the PARP-inhibitor Olaparib. A synergistic effect on cell death was evident when these drugs were used concurrently. The improved SN38/irinotecan efficacy was accompanied by the down-regulation of DNA repair genes. Thus, differential methylation of *MCC* is potentially a valuable biomarker to identify colorectal cancers suitable for irinotecan therapy, possibly in combination with PARP inhibitors.

## 1. Introduction

Colorectal cancer is the second leading cause of cancer-related mortality in the world (https://gco.iarc.fr/today (accessed on 25 May 2022)). Immunotherapy or molecular targeted therapies are available for a subset of patients, but 5-fluorouracil-based chemotherapy is still a mainstay of treatment for advanced cancer, usually administered with folinic acid and oxaliplatin (FOLFOX), or with irinotecan and leucovorin (FOLFIRI). Despite intensive research, relatively few predictive biomarkers are in routine use for evaluating responsiveness to the various chemotherapy regimens. 

The ‘Mutated in Colorectal Cancer’ (*MCC*) gene was discovered due to its close proximity to the *APC* gene on chromosome 5 [1], but it has *APC*-independent roles in colorectal cancer. MCC is emerging as a tumor suppressor involved in at least two cellular processes, the DNA damage response and cell–cell adhesion [2,3,4,5,6,7,8,9,10]. We showed that CpG island hypermethylation is a common cause of *MCC* silencing in serrated polyps and carcinomas in the colon [5,11]. *MCC*-methylated tumors are associated with poorly differentiated, circumferential, and mucinous tumors, as well as increasing T stage, larger tumor size, proximal colon location [2], and down-regulation of *MSH3* gene expression [11]. *MCC*-methylated cancers include CpG island methylator phenotype (CIMP)-positive tumors [5,11] that are potentially more responsive to irinotecan [12]. These findings have raised the prospect of exploiting *MCC* silencing in cancer therapy, and particularly in relation to irinotecan responsiveness [3].

Irinotecan is mainly used to treat stage IV colorectal cancers, but would also be potentially effective in stage III cancers that are CIMP-positive [12,13]. CIMP is characterized by concordant promoter hypermethylation and silencing of multiple tumor suppressor genes, and is identified by a PCR-based 5-gene marker panel. CIMP-high (H) is defined when at least 3/5 markers are positive [14]. It overlaps with the high microsatellite instability (MSI-H) phenotype, which includes DNA mismatch repair deficient cancers, and is mainly caused by the silencing of the *MLH1* gene in sporadic colorectal tumors. Neither CIMP-H or MSI-H cancers are responsive to standard 5-fluorouracil-based chemotherapy [15]. Therefore, investigating CIMP-related methylation biomarkers may help to optimize patient selection for irinotecan-based chemotherapy. 

Here, we show that *MCC* hypermethylation, rather than copy number variation (CNV), is the driver of *MCC* loss in CIMP subtypes. We also show that in addition to promoter hypermethylation, *MCC* silencing is associated with hypomethylation of distant gene regions in a small subset of colorectal cancers. Furthermore, *MCC* deficiency increases the efficacy of both irinotecan and PARP inhibitor Olaparib in two cell line models and causes synergy when they are used concurrently. 

## 2. Materials and Methods

### 2.1. Analysis of The Cancer Genome Atlas (TCGA) Datasets

The 2012 TCGA cohort of 276 colorectal carcinomas was accessed through the cBioPortal for Cancer Genomics platform (https://www.cbioportal.org (accessed on 9 November 2021)) [16,17], and sample annotations were accessed from the associated publication [18]. The TCGA PanCancer Atlas dataset that contains comprehensive integrated molecular analyses for 594 colorectal carcinomas [19] was obtained using the SMART App (http://www.bioinfo-zs.com/smartapp (accessed on 19 August 2021)) [20]. Matching *MCC* copy number variation, differential methylation, mutation, and expression data were available for 271–285 colon cancers (COAD) and 86–91 rectal cancers (READ).

### 2.2. Cell Lines

HCT116 colon cancer cell line was obtained from the American Type Culture Collection (ATCC CCL-247, Manassas, VA, USA), and was maintained at 37 °C in 5% CO_2_ in McCoys Medium Modified (Catalog No. 16600082, Thermofisher Scientific, Waltham, MA, USA) supplemented with 10% fetal bovine serum and penicillin/streptomycin. Genomic sequence spanning exons 2–6 of the *MCC*-201 isoform ENST00000302475.8 (corresponding to exons 4–8 of *MCC*-202 isoform) was deleted using a CRISPR-Cas9 mediated approach with two guide RNAs targeting *MCC* sequences GCAGCCCTGGCATCACTAAAGGG and CAGACAGTCGAGGAGATTGAGGG. The loss of *MCC* protein expression was verified by Western blotting. A total of six clonal *MCC*-deleted HCT116 cell lines were pooled after their first few passages, and then split into seven biological replicates. All replicates were maintained independently, and used in the experiments with six biological replicates of the parental cell line as unmodified controls (all used at passage > 20). *MCC* knockdown in HCT116 cells was carried out as previously described [2,21]. All modified cell lines were verified by DNA fingerprinting at the time of the experiments at Garvan Molecular Genetics Facility, Garvan Institute of Medical Research. *MCC*-deleted and *MCC*-WT mouse embryo fibroblasts (MEF) were raised as previously described [3].

### 2.3. Cell Proliferation

*MCC*-expressing and *MCC*-deficient HCT116 cells were seeded in a 24-well plate at low density (~10% confluency). The plate was placed in the IncuCyte Zoom (Sartorius). Cell confluency was recorded in real-time. A total of nine images per well were acquired every 2 h. The average confluency of all nine images of each scan was determined. A percentage confluency vs. time growth curve was plotted for each of the cell types. 

### 2.4. Cell Viability Assay

SN38 and Olaparib (AZD2281) were purchased from Selleckchem (Houston, TX, USA). SN38 and Olaparib were dissolved in dimethyl sulfoxide at a concentration of 10 mmol/L, and stored at −20 °C until the in vitro experiment. Cell viability assay was performed using resazurin-based cell viability reagent Alamar Blue, following the kit protocol (Cat No. DAL1025; Thermofisher Scientific, Waltham, MA, USA). Cells were treated in their log phase with an increasing concentration of SN38 (10 nM to 100 nM) and Olaparib (1–350 nM) in 96-well tissue culture plates for 48 hr. Plates were read (excitation, 530–560 nm; emission, 590 nm) using a 96-well plate reader (Spectramax iD5; Molecular Devices; San Jose, CA, USA), and the percentage of surviving cells relative to untreated control was measured.

The IC50 of *MCC*-knockdown HCT116 cells was determined using the CellTitre 96 AQueous Non-Radioactive Cell Proliferation Assay (Promega, Madison, WI, USA). Cells were treated with a range of concentrations of SN38 (1 nM–100 μM). The plates were read at an absorbance of 490 nm on the FLUOstar Omega Microplate Reader (BMG Labtech, Ortenberg, Germany). IC50 values were calculated using GraphPad Prism. 

### 2.5. Drug Synergy Experimental Design

The drug synergy experiment was performed based on the combination index (CI) method using CompuSyn software Ver 2.0 (Compusyn, INC. PD Science LLC; New York, NY, USA) [22,23]. Based on the IC50 of each drug, six drug combinations (IC50 multiplied by 0.25, 0.5, 1, 2, 4, and 8) were tested to determine the dose-effect curve of SN38 and Olaparib, respectively, in five biological replicates of *MCC*-KO and *MCC*-WT cell lines. Regarding the optimal combination ratio for maximal synergy, the IC50 considered for SN38 and Olaparib was 1 nM and 10 nM, respectively. The drug treatment experiments were repeated at least three times.

### 2.6. Western Blot Analysis

HCT116 cells were grown in 10 cm tissue culture plates for 24 h (exponential growth), treated with the drugs, and harvested by scraping 24 h post treatment. Cells were centrifuged at 1500× *g* rpm for 5 min, and pellets were washed in cold Dulbecco’s phosphate-buffered saline (DPBS). Pellets were then dissolved in radioimmunoprecipitation assay (RIPA) buffer (Sigma-Aldrich, MO, USA) supplemented with Pierce Protease and Phosphatase Inhibitor Mini Tablet (Cat No. A32959; Thermofisher Scientific, Waltham, MA, USA) for whole cell lysates. Cell lysates containing equal amounts of protein were separated by SDS-PAGE, and transferred to polyvinylidene difluoride membrane under the appropriate conditions. 

The following antibodies were used: total DNA-PKc, PARP, β-Actin (Cat No. 12311, 9542, 8457, respectively; Cell Signalling Technology, Danvers, MA, USA), MCC (Cat No. 610740; BD Biosciences, Franklin Lakes, NJ, USA), Phospho-DNA-PKc Ser2056 (Cat No. 68716; Cell Signalling Technology, Danvers, MA, USA), and ATR (Cat No. sc-515173; Santa Cruz Biotechnology, Dallas, TX, USA). 

Bands were visualized by enhanced chemiluminescence (ECL) horseradish peroxidase substrate (Western Lightning Plus ECL, PerkinElmer, Waltham, MA, USA). Each experiment was repeated at least three times. Blots were quantified using Image Lab v5.2.1 image analysis software (Bio-Rad Laboratories, Hercules, CA, USA). PARP, cleaved PARP, DNA-PKc, pDNA-PKc, and RAD51 levels were normalized to β-Actin, and subsequently the ratio of pDNA-PKc/DNA-PKc was determined. The entire Western blots can be found in the Appendix A.

### 2.7. PARP Immunofluorescence

Immunofluorescence analysis was performed as previously described [2]. Briefly, *MCC*-expressing and *MCC* knockdown cells were fixed with 4% paraformaldehyde for 20 min, permeabilized with 0.1% Triton X-100 for 20 min, blocked with 10% FBS for 30 min, and incubated with primary antibodies MCC (BD, 610740) and PARP (Cell Signaling, 9542) overnight at 4 °C. The signal was detected using conjugated secondary antibodies Alexa Fluor 488 and Alexa Fluor 647 (Jackson ImmunoResearch Laboratories, West Grove, PA, USA) for 20 min at RT, followed by DAPI to visualize the nuclei (Sigma, Saint Louis, MO, USA) for 5 min, before mounting with Vectashield (Vector Labs, Newark, CA, USA) on glass slides. Images were acquired using a Leica DMI 6000 SP8 confocal microscope.

### 2.8. qPCR Analysis

cDNA was prepared using the Quantitect Reverse Transcription Kit (205311; Qiagen, Hilden, Germany). Expression of DNA damage response genes was analyzed in HCT116 cells, treated with 20 nM SN38 for 20 h. A total of six clonal *MCC*-deleted HCT116 cell lines were pooled and maintained independently in culture as six biological replicates to use for the experiment. The qPCR was conducted in triplicate for each specimen. The following TaqMan assays (Life Technologies, Carlsbad, CA, USA) were used: Hs99999905_m1 (PARP1), Hs00947967_m1 (RAD51), Hs00992123_m1 (ATR), Hs99999905_m1 (GAPDH).

### 2.9. Animal Experiments

All mouse experiments were approved by the Garvan and St Vincent’s Animal Ethics Committee. Athymic BALB/c female nude mice were supplied by Australian BioResources (Moss Vale, Australia). *MCC*-expressing and *MCC*-knockdown HCT-116 cells (5 × 10^6^) were resuspended in 100 µL PBS containing 0.2% FBS and injected into the left or right flank of three mice per respective group. Mice were treated with 30 mg/kg irinotecan hydrochloride in DMSO (2% *w*/*v*) on days 1, 5, and 10. Tumor volume was measured with a caliper using the formula: tumor volume (mm^3^) = (length × width × width)/2. Tumor retention and growth was assessed by injecting 10 mg/mL D-luciferin intraperitoneally at 10 μL per gram body weight, and imaged under anesthesia on the IVIS Spectrum In Vivo Imaging System (PerkinElmer, Waltham, MA, USA). 

### 2.10. Statistical Analysis

Differences in protein levels were compared with nested or ordinary one-way ANOVA or the Kruskal–Wallis test, and mRNA expression levels with *t*-tests, ANOVA, or the corresponding non-parametric tests. The level for statistical significance was set at ≤0.05. Association between gene expression and differential methylation was compared with a Mann–Whitney test (individual CpG sites) or with a Kruskal–Wallis test (regional methylation patterns). Association between methylation clusters (CIMP-H, CIMP-L, Cluster 3, Cluster 4) and *MCC* methylation was determined with a Kruskal–Wallis test. Contingency analysis of change in CNV in association with methylation clusters was performed with a two-sided Chi squared test, comparing diploid/gain versus gene deletion (shallow or deep).

## 3. Results

### 3.1. Differentially Methylated Genomic Regions Can Identify MCC-Deficient Tumors

Since CIMP-H is reported to sensitize colorectal tumors to irinotecan, we examined the mechanisms by which *MCC* is lost in those cancers to facilitate potential biomarker development. We previously showed that CpG island hypermethylation is a cause of *MCC* gene silencing, and can be detected with methylation-specific PCR [2,5]. Here, we focused on gene-wide methylation patterns and copy number alterations from genomic data that allow for more extensive analysis. While *MCC* promoter methylation is found in almost all CIMP-H colorectal cancers, we set out to understand how this relates to the level of methylation, using the HM27 array data available for the TCGA 2012 cohort [18]. All CIMP-H and around half of CIMP-low (L) cancers showed *MCC* methylation beta-value > 0.5, indicating strong methylation, while the non-CIMP methylation clusters 3 and 4 showed gradually decreasing levels of beta-values (Figure 1A). Most CIMP-H and CIMP-L cancers had *MCC* diploid status (84–93%), while the two non-CIMP methylation clusters included a higher proportion of cancers with *MCC* deletions (24–28%) (Figure 1B).

We next analyzed the TCGA 2018 cohort [16,17,19,20], where matching transcriptome and HM450 methylome data were available from 271 colon and 86 rectal cancers. As shown for other genes in colorectal cancer, the hypermethylated CpG island of *MCC* is surrounded by differentially methylated regions, known as CpG shores and shelves [24,25]. We focused on the *MCC*-201 (ENST00000302475.8) transcript that is the predominant isoform in the colon and rectum. Hypermethylation of multiple individual CpG sites (beta-value > 0.5) was associated with *MCC*-201 mRNA down-regulation in the colon, including all three HM450-targeted sites in the CpG island and six of nine sites in the shores (Figure 1C; Appendix A). Notably, two CpG sites (cg23958684, cg06628473) in the S-shore and S-shelf were hypomethylated (beta-value < 0.4), which was also strongly associated with *MCC*-201 down-regulation. 

There was some variation between colon and rectal cancers in the number of hypermethylated CpG sites throughout the gene (Appendix A). Therefore, we arranged the cancers into groups according to the co-occurrence of differentially methylated regions (Appendix A). This showed that CpG island hypermethylation is usually accompanied by differential methylation of the shores or shelves in the same tumors. CpG shore hypermethylation can also occur independently, but this is not associated with low *MCC* expression (Appendix A; Figure 1D,E). In contrast, S-shore/shelf hypomethylation is strongly associated with *MCC* down-regulation, even in the absence of hypermethylation, in both colon and rectal cancers (indicated with blue dots in Figure 1D,E). 

Since CIMP-H is strongly associated with MCC diploid status, we next investigated how the regional methylation patterns of *MCC* correlate with its CNV status. Here, we used a more stringent beta-value > 0.6 for each CpG site as a threshold for *MCC* hypermethylation (Figure 1F,G). In the colon, 18% (50/272) of cancers had strong *MCC* CpG island hypermethylation, and these were diploid or showed copy number gain (Figure 1F). This is similar to CIMP-H, which has an inverse correlation with loss of heterozygosity in key tumor suppressor genes [26]. In the rectum, there was no difference in the methylation patterns between diploid and CNV cancers (Figure 1G). The CpG island hypermethylation frequency was 7% (6/91), and was almost always accompanied by shore/shelf hypomethylation in rectal tumors. Taken together, the genomic data from the two TCGA colorectal cancer cohorts suggest that *MCC* is silenced by CpG island hypermethylation in CIMP-H colon cancers, while low *MCC* expression is associated with other factors in non-CIMP cancers, such as shore/shelf hypomethylation and gene deletions. 

### 3.2. MCC Knockdown Sensitises Colon Cancer Cells to SN38/Irinotecan-Induced Cell Death

*MCC* gene knockdown in tumor cells leads to increased DNA breaks and cell cycle perturbation after exposure to DNA damaging agents [3,8]. To investigate the effect of *MCC* deletion on SN38/irinotecan treatment, we tested MEFs isolated from *MCC*-knockout (KO) mice and their wild type (WT) siblings. We also examined HCT116 colon cancer cells that were beta-catenin-mutated and CIMP-positive but had endogenous *MCC* expression. *MCC* knockdown in HCT116 cells caused a significant increase in cell proliferation (*p* < 0.0001) (Figure 2A). When exposed to rising concentrations of SN38, *MCC*-knockdown or deletion increased cell death, and caused a substantial reduction in IC50 value in both HCT116 cells and MEFs (Figure 2B,C). 

A xenograft model of HCT116 cells was established to determine the effect of *MCC* knockdown on irinotecan response in vivo. Athymic BALB/c nude mice were injected with non-targeted (NT, vector-only control) or *MCC*-knockdown HCT116 cells. When tumors reached 200 mm^3^, the mice were injected intraperitoneally with 30 mg/kg irinotecan hydrochloride in vehicle (2% *w*/*v* DMSO). In a parallel experiment, the tumors were allowed to grow without any treatment. The *MCC*-deficient tumors grew significantly faster than *MCC*-expressing tumors. After irinotecan treatment, tumor growth stabilized at 300 mm^3^ on day 18, and then started to decline faster for *MCC*-deficient than *MCC* expressing cells (*p* < 0.05) (Figure 2D). 

### 3.3. MCC Knockdown Induces PARP Expression in Colon Cancer Cells In Vivo

PARP proteins are important nuclear sensors for DNA damage, and mediate the repair of DNA breaks through the non-homologous end-joining (NHEJ) and base excision repair (BER) pathways. In our previous study, we showed that *MCC* deletion or knockdown exacerbates H_2_O_2_ or SN38-generated DNA damage, as shown by increased H2AX protein expression or comet assay [3]. Here, we analyzed the effect of *MCC* knockdown or deletion on PARP expression after SN38/irinotecan exposure. PARP expression was higher in *MCC*-deficient xenograft tumors, regardless of irinotecan treatment (Figure 2E). Irinotecan exposure in vivo induced *MCC* expression both in vector control cells and in *MCC*-knockdown cells. The latter had residual *MCC* expression due to incomplete knockdown. Longer exposure of the Western blot revealed that the upregulated protein was most likely the phosphorylated form, higher molecular weight MCC. In a separate in vitro experiment, we treated HCT116 cells with 1 μM of SN38 for 2 h. Immunofluorescence analysis revealed increased nuclear localization of PARP in *MCC*-knockdown cells (Appendix A).

### 3.4. MCC Deletion Alters the Transcriptional Response to SN38-Induced DNA Damage

We then analyzed PARP expression in HCT116 cells with CRISPR-mediated complete deletion of *MCC*. Here, basal expression of PARP protein was also increased in *MCC*-deleted cells in vitro (Figure 3A). The basal levels of other selected DNA repair proteins were similar (ATR, RAD51) or slightly increased (DNA-PKc) in *MCC*-deleted HCT116 cells (Figure 3A; Appendix A). SN38 exposure boosted phosphorylation of DNA-PKc (S2056) in both *MCC*-WT and *MCC*-KO cells. 

Basal expression of *PARP* was also upregulated at the mRNA level in *MCC*-deleted HCT116 cells (Figure 3B). Similar upregulation was observed for *ATR* and *RAD51*. The transcription of all three genes was downregulated following SN38 treatment in *MCC*-KO, but not in *MCC*-WT cells. This is consistent with our previous data in the *Mcc*-deleted mouse colon and MEF cells, where multiple DNA repair genes were downregulated in response to the generation of DNA damage [3]. 

### 3.5. PARP Inhibitors Synergise with SN38 in MCC-Deleted Cells

Tumors rely on PARP-mediated DNA repair for survival, and are sensitive to its inhibition. If tumors are defective for a complementary DNA repair pathway, the therapy accelerates cancer cell death through synthetic lethality, which is the rationale for using PARP inhibitors (PARPi) to treat BRCA-defective cancers [27]. Due to the increased PARP expression in *MCC*-deficient cells, we hypothesized that *MCC* loss may enhance PARPi sensitivity. 

HCT116 and MEF cells were given rising concentrations of Olaparib for 20 h, and IC50 was quantified using the Alamar blue assay. *MCC* deletion caused a 1000-fold decrease in the IC50 value of Olaparib in HCT116 cells, and a small decrease in MEFs (Figure 3C,D). Thus, *MCC* deletion sensitizes HCT116 cancer cells or MEFs to cell death in response to either SN38 or PARPi. 

For the MEFs, we then tested a 1 nM concentration of SN38 with variable concentrations (0–100 nM) of Olaparib (Figure 3E). In *Mcc*-WT MEFs, this resulted in only up to 20% cell death, while there was a clear dose response in *Mcc*-KO MEFs, and the highest concentration of Olaparib tested caused 80% cell death after 20 h. Drug synergy was systematically tested with rising concentrations of both drugs in HCT116 cells. The optimal combination ratio for maximal synergy was close to 1 in 10, based on the IC50 values for SN38 and Olaparib in *MCC*-KO cells, respectively. The highest efficacy for the drugs was ~60% cell death in *MCC*-WT cells, and 90% in *MCC*-KO cells (Figure 3F). There was a weak additive effect of the drug combination in *MCC*-WT cells but a clear synergistic effect in *MCC*-KO cells.

## 4. Discussion

CIMP-H represents a clinically relevant phenotype resulting from multiple tumor suppressor genes that are silenced by hypermethylation. Our analysis of the TCGA cohorts shows that *MCC* is highly methylated in all CIMP-H and half of CIMP-L colorectal cancers. Apart from CpG island hypermethylation, *MCC* shore/shelf hypomethylation also correlates strongly with low gene expression. Hypomethylation is not known to cause gene silencing directly. It is possible that *MCC* gene hypomethylation is associated with additional factors that regulate gene expression. Other factors that cause loss of gene expression have been previously reported for *MCC*, such as microRNA targeting in colon and liver cancer cells [28,29,30]. Moreover, LINE-1 retrotransposon insertion in germline DNA and a lack of MCC protein expression in normal tissue have been reported in a subset of liver cancer patients [31]. This indicates that *MCC* expression levels can also vary in normal tissue due to genetic variation.

Our study suggests that the loss of *MCC* expression, an individual gene strongly associated with CIMP, increases tumor sensitivity to irinotecan and PARPi separately, and even further in combination. FOLFIRI is one of the standard first-line therapies in metastatic colorectal cancer. PARPi are approved to treat BRCA-defective ovarian, breast, and prostate cancers, but are not yet approved for colorectal cancers. A total of 13% of non-MSI-H colorectal cancer cell lines were found to be highly sensitive to Olaparib [32]. Furthermore, a synergistic effect of SN38 with veliparib, Olaparib, or rucaparib was demonstrated in several colorectal cell lines, independent of MSI-H status [33,34,35]. Furthermore, the manipulation of several genes was shown to increase or decrease SN38/PARPi sensitivity in HCT116 cells [33]. However, a Phase 2 randomized trial of veliparib and FOLFIRI combination therapy did not show increased efficacy compared to standard FOLFIRI treatment in metastatic colorectal cancers [36]. This is possibly due to the lack of patient selection for predictive markers. 

Previous studies suggest that MCC has a role in the cellular DNA damage response that is relevant for cytotoxic drug efficacy. MCC deficiency increases DNA breaks in response to irinotecan in colon cancer cells, as well as in response to free radical generation by H_2_O_2_ in mouse embryo fibroblasts [3]. The MCC protein localizes to the nucleus, and is phosphorylated after radiation exposure, and ectopic *MCC* expression slows down cancer cell proliferation [8]. Several single nucleotide polymorphisms (SNPs) within the *MCC* gene correlate with sensitivity to the cytotoxic drug cytarabine in acute myeloid leukemia patients [37]. Furthermore, *MCC* expression is induced by cytarabine exposure in lymphoblastoid cell lines [37].

Here, we found that the increased drug efficacy is accompanied by down-regulation of DNA repair genes after induction of DNA damage. This is consistent with our previous data on the *Mcc*-∆IEC mice, which showed the down-regulation of the cell cycle and DNA damage response pathways in the inflamed colon [3]. These included targets of two major transcription factors, E2F and MYC. The MCC protein may support the transcription of multiple key genes across several DNA repair pathways after the induction of DNA damage. Therefore, when MCC activity is absent, cancer cells can accumulate DNA breaks which make them more sensitive to irinotecan and PARP-induced cell death. It is not clear what activates the nuclear localization and DNA repair function of MCC, nor how MCC supports transcription. We previously showed that there are several candidate ATM/ATR/DNA-PK phosphosites in the MCC protein, which is phosphorylated in response to UV radiation [8]. The exact mechanism and role of MCC in the DNA damage response network remains to be elucidated, but may involve the regulation of its DNA repair activity and nuclear localization through phosphorylation changes.

## 5. Conclusions

In conclusion, reduced *MCC* expression sensitizes mouse embryo fibroblast and HCT116 colon cancer cells to SN38/irinotecan-induced cell death, and PARP inhibitor Olaparib augments this effect. If these results can be confirmed in further cancer cell lines, *MCC* alterations should be further evaluated for patient management. Differential methylation of *MCC* is potentially a valuable biomarker to identify colorectal cancers suitable for irinotecan therapy, possibly in combination with PARP inhibitors.

## Figures and Tables

**Figure 1 cancers-14-02859-f001:**
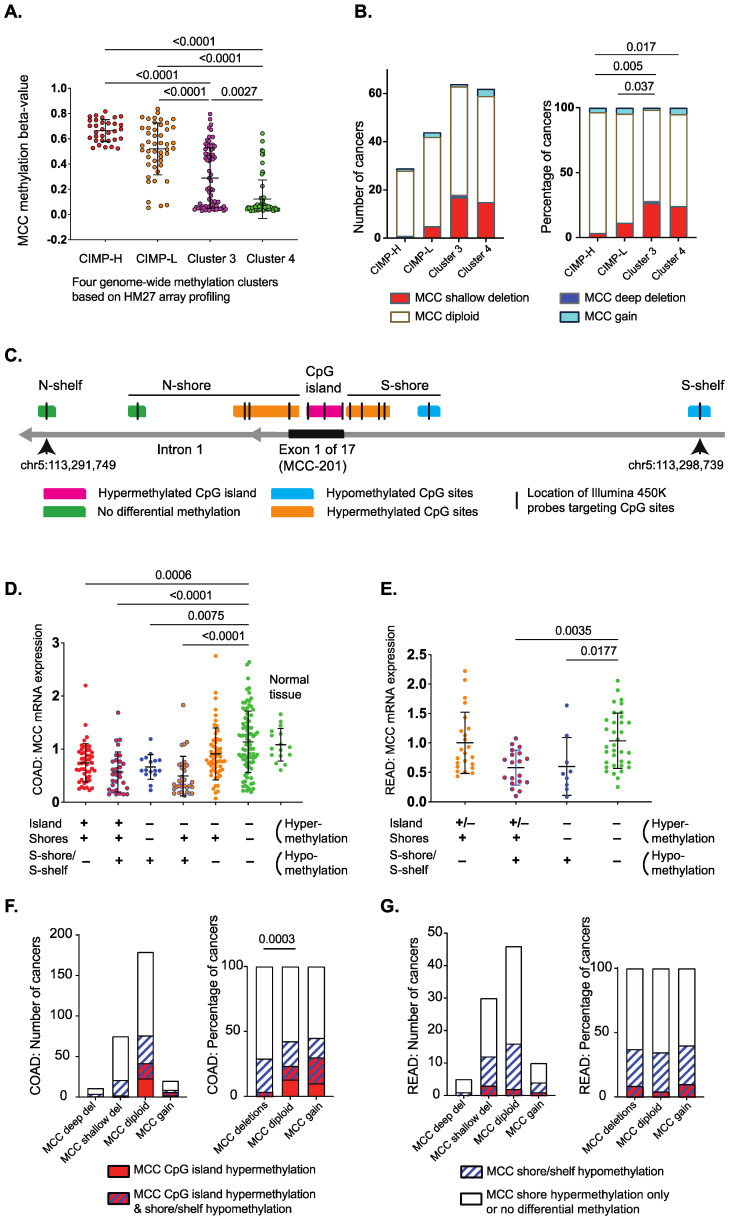
Genomic data from TCGA colorectal carcinomas shows *MCC* gene down-regulation is associated with hypermethylation of the CpG island or hypomethylation of the S-shelf/shore (data from cBioPortal and SMART App) [16,17,19,20].(**A**) Methylation of *MCC* (HM27 array profiled) within methylation clusters (CIMP-H, CIMP-L, Cluster 3, Cluster 4) of colorectal carcinoma [18]. Statistical significance was determined using one-way ANOVA with a Kruskal–Wallis test. (**B**) Copy number variation of *MCC* (HM27 array profiled) within methylation clusters (CIMP-H, CIMP-L, Cluster 3, Cluster 4) of colorectal carcinoma [18]. Statistical significance was determined via a Chi squared test of MCC diploid/gain status vs. gene deletions. Data were obtained from cBioPortal [16,17].(**C**) Location of differentially methylated CpG sites in the CpG island, shores, and shelves of the MCC-201 isoform in colon cancer (HM450 arrays, TCGA 2018). Methylation data were obtained using the SMART App [20]. Rectal cancers show fewer hypermethylated CpG sites (details in Appendix A). The genomic coordinates and location of the features were obtained from UCSC Genome Browser GRCh38/hg38 Assembly (December 2013). (**D**) *MCC* mRNA down-regulation is associated with differential methylation of the CpG island, shores, and shelves in colon cancer (TCGA 2018 COAD). ‘+’ refers to the presence of hypermethylation or hypomethylation and ‘−‘ refers to the absence of hypermethylation or hypomethylation. Detailed data are shown in Appendix A. Statistical significance was determined using the Kruskal–Wallis test. Error bars show mean ± SD. Methylation and gene expression data were obtained using the SMART App [20]. (**E**) *MCC* mRNA down-regulation is associated with differential methylation of the CpG island, shores, and shelves in rectal cancer (TCGA 2018 READ). ‘+’ refers to the presence of hypermethylation or hypomethylation and ‘–‘ refers to the absence of hypermethylation or hypomethylation. Detailed data are shown in Appendix A. Statistical significance was determined using the Kruskal–Wallis test. Error bars show mean ± SD. (**F**) *MCC* CpG island hypermethylation is associated with diploid or copy number gain status in colon cancer, while shore/shelf hypomethylation is evenly distributed in diploid and CNV cancers. The beta-value thresholds were >0.6 (hypermethylation) and <0.4 (hypomethylation). Cancers that display no differential methylation or only shore hypermethylation were combined as one group. Methylation and CNV data were obtained using the SMART App [20]. (**G**) Strong *MCC* CpG island hypermethylation is rare in rectal cancer, while shore/shelf hypomethylation is evenly distributed in diploid and CNV cancers. The beta-value thresholds were >0.6 (hypermethylation) and <0.4 (hypomethylation). Cancers that display no differential methylation or only shore hypermethylation were combined as one group.

**Figure 2 cancers-14-02859-f002:**
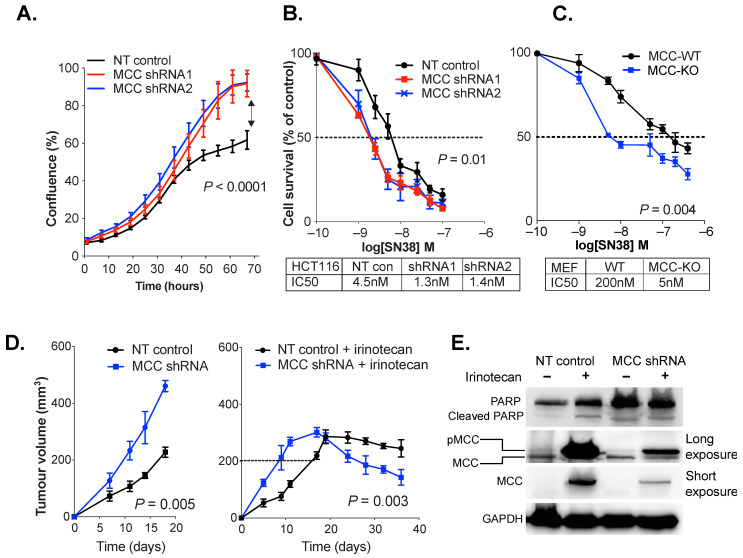
*MCC* deficiency increases DNA damage, PARP nuclear localization and cell death in response to SN38/irinotecan exposure. (**A**) *MCC* knockdown (shRNA1 and shRNA2) increases rate of HCT116 cell proliferation in vitro (IncuCyte). Statistical significance was determined using a two-way ordinary ANOVA. Error bars show mean ± SD of 3 replicates. (**B**,**C**) *MCC* knockdown or deletion sensitises HCT116 cells and MEFs to SN38 in vitro. Cells were treated with rising concentrations of SN38 (1 nM to 1μM), and harvested after 48 h. IC50 was extrapolated from log-dose vs. response curves using GraphPad Prism. Statistical significance was determined using a paired *t*-test. Error bars show mean ± SD of 5 replicates. (**D**) *MCC*-deficient tumors grow significantly faster and are more responsive to irinotecan treatment than *MCC*-expressing tumors. Athymic BALB/c nude mice were injected with non-targeted (NT) HCT116 control cells or *MCC*-shRNA2 cells. When tumors reached 200 mm^3^, half of the mice received 3 doses of 30 mg/kg irinotecan hydrochloride (right) on days 1, 5 and 10. Half of the mice received no treatment (left). Statistical significance was tested using two-way ANOVA (left panel) and a paired *t*-test (last four time points in right panel). Error bars show mean ± SEM. (**E**) Xenograft-harvested *MCC*-shRNA and NT tumors show increased MCC phosphorylation in response to irinotecan, and *MCC*-shRNA tumors show increased PARP expression regardless of treatment. Protein lysates were extracted by RIPA buffer and 30 μg of protein per sample was analyzed. The Western blot film was developed at low exposure (2 s) and long exposure (120 s).

**Figure 3 cancers-14-02859-f003:**
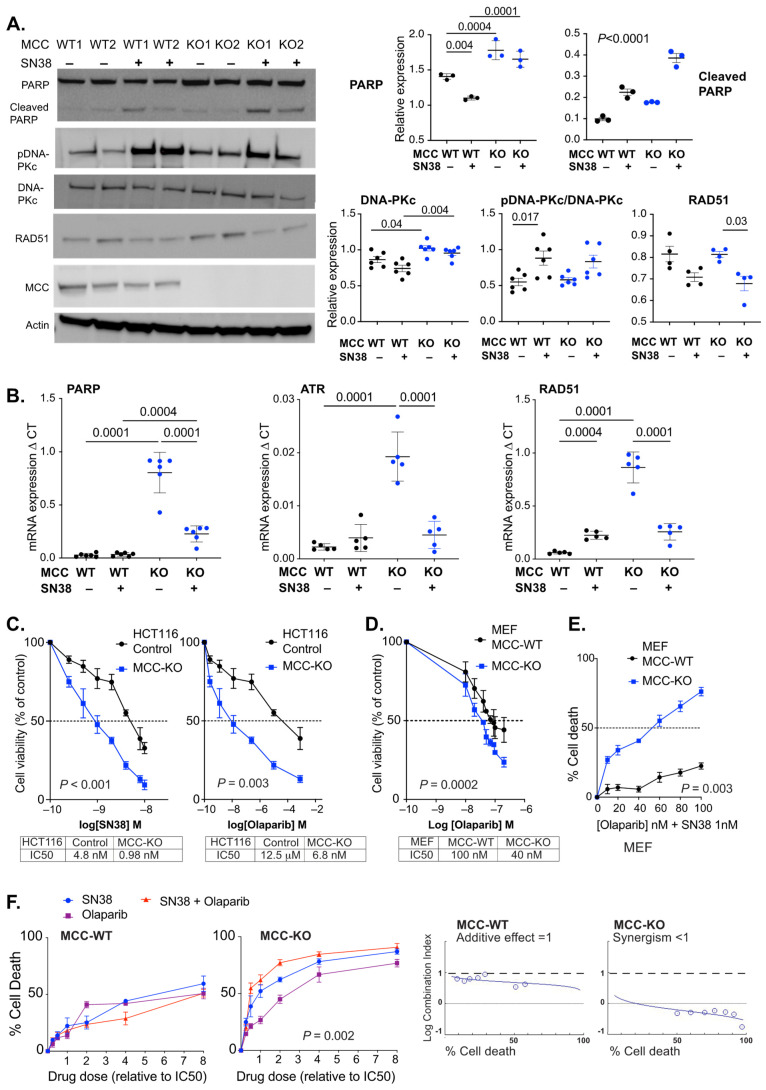
*MCC* deletion sensitizes HCT116 cells to SN38 and Olaparib. (**A**) *MCC* deletion increases PARP protein expression. *MCC*-WT and *MCC*-KO cells were cultured with or without 20 nM SN38 for 20 h. A representative Western blot (left) and quantification of protein expression (right). Error bars show mean ± SD of three experiments with three biological replicates for PARP, and of one representative experiment with three–six biological replicates for the others. Statistical significance was determined using ordinary or nested one-way ANOVA and the Kruskal–Wallis test (if <5 replicates). PARP cleavage indicates cells undergoing apoptosis. (**B**) *MCC*-KO HCT116 cells show increased mRNA expression of DNA repair genes *PARP1*, *RAD51* and *ATR* after 20 h of culture, which is reversed with SN38 exposure. Cells were treated with 20 nM SN38 for 20 h. Statistical significance was determined using one-way ANOVA. Error bars show mean ± SD of five–six biological replicate cell lines. (**C**,**D**) *MCC*-KO HCT116 cells and MEFs were exposed to rising concentrations of SN38 (0.25 nM to 10 nM) and Olaparib (2.5 to 80 nM), and cells were harvested after 20 h. IC50 was calculated from log-dose vs. response curves generated in Graphpad Prism. Statistical significance was determined using a paired *t*-test. Error bars show mean ± SD of five biological replicates. (**E**) *MCC*-deletion enhances the sensitivity of MEF cells to a combination treatment with SN38 (1 nM) and Olaparib (0–100 nM). Statistical significance was determined using a paired *t*-test. Error bars show mean ± SD of three biological replicates. (**F**) Strong drug synergy is observed with SN38/Olaparib combination treatment (red) in *MCC*-KO HCT116 cells. Cells were treated with increasing doses of drugs (multiples of IC50 dose of each drug). Statistical significance was determined using one-way repeated measures ANOVA (drug doses 0.5–4). Error bars show mean ± SD of five biological replicates. Graphic output is obtained from CompuSyn Report.

## Data Availability

Publicly available TCGA datasets were analyzed in this study. The data for *MCC* mRNA expression, CNV, and methylation in colorectal tumors can be found at www.cbioportal.org/results/plots?cancer_study_list=coadread_tcga (accessed on 9 November 2021) and www.bioinfo-zs.com/smartapp (accessed 19 August 2021).

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
