# Peer review of "MCC Gene Silencing Is a CpG Island Methylator Phenotype-Associated Factor That Predisposes Colon Cancer Cells to Irinotecan and Olaparib"

_cancers, 2022, doi:10.3390/cancers14122859_

Round 1

Reviewer 1 Report

The authors revised the manuscript in detail following the Reviewer's suggestions and also answered to all questions. 

In this new version they discuss and comment all the presented results without leaving anything unexplained.

This version is for me acceptable for publication.

However I signal the authors to check Figure 2A, there is something strange in the title of y axis.

Reviewer 2 Report

The authors have addressed many of the reviewers’ comments and questions. Performance of the study in only a single colon cancer cell line is a major limitation of the work overall, however the findings reported in the manuscript are interesting and worthy of publication.

1.       1. Line 46: please explain abbreviation DPBS

2.       2. Line 51: ‘difluoride’ (not difluoridem)

This manuscript is a resubmission of an earlier submission. The following is a list of the peer review reports and author responses from that submission.

Round 1

Reviewer 1 Report

This interesting manuscript has used database analysis and a cell line model to further characterise MCC function in colorectal cancer cells. In general, the study is well-structured, English language and manuscript writing are excellent and the quality and presentation of figures are very good. The study builds upon previous work of the investigators and adds new data to the field. A major limitation of the work is the use of a single colorectal cancer cell line (HCT116) for experimental studies, and it is feasible that this may restrict the broader application of results of the work. Overall, I recommend the study for publication pending consideration of the following comments and questions. 

  1. In the Abstract (line 28), the authors state that SN38/irinotecan treatment increases DNA damage. Where are these data – they do not seem to be included in the figures or text (is it possible that some of the files have not uploaded?).
  2. In Figure 1B, have the authors included data for ‘MCC deep deletion’, or are the values so low that they not visible in the bar graph?
  3. The authors seem to have used a confluency assay to estimate cell proliferation in Figure 2A, but then in figures 2B and 2C where they are estimating IC50 values, they use Alamar Blue. Is there any reason for using 2 different assays? Does MCC knockdown alter cell size, cell spread on the culture surface or contact inhibition?
  4. The resolution of images in Figure 2F is not suitable for publication. Images are out of focus and it is not possible to discern the nuclear or cytoplasmic localisation of PARP that the authors mention in the text. Could these images please be replaced with images that are in focus – a slightly higher magnification may also help.
  5. In Figure 3A, the authors show Western blot images for DNA-PKc and pDNA-PKc, but have only included the blot for pATR. Please add the matching blot for ATR as results are difficult to interpret without this.
  6. I am not sure that I understand how the authors are quantitating their Western blots. It is usual to normalise each band to its respective b-actin. However, to understand whether a treatment is inducing phosphorylation of a particular protein, the (b-actin normalised) phosphorylated protein level is divided by the (b-actin normalised) level of the corresponding total protein. Could the authors please clarify that this is what they have done as it is not clear from the figure legends or Methods section (Figure 3).
  7. I feel that the limitations of the study and the limitations of the experimental design have not been clearly enunciated and that these should be added to the manuscript. In particular, use of a single cell line is a major limitation of the study that needs to be clearly defined. While creation of biological replicates enables additional points to be added to graphs, it does not diminish the limitation of sole use of one extensively cultured cell line with its single genetic background. Reasons for the lack of verification of any results with additional cell lines should also be added. The amount of time (number of passages) that control and MCC-deleted sublines were cultured should also be added to the Methods section.
  8. The number of replicates used for experiments and the number of times that experiments were repeated are difficult to follow. For example, the authors state that they only used 3 mice per group, however I cannot find a statement detailing how many times experiments were repeated. In addition, I cannot find descriptions regarding how consistent results from the biological replicates were. Was this the reason why all biological replicates were not used in experiments?
  9. In this study, the authors have made several interesting observations regarding MCC downregulation and irinotecan/PARP inhibitor sensitivity. However, I feel that the conclusions of the authors may be overstated at this time. Firstly, I don’t feel that MCC methylation profiles have been evaluated sufficiently/specifically by the authors in relation to their potential use as a marker of CIMP. In addition, HCT116, the only colon cancer cell lines used in the study, carries a particular set of mutations and has microsatellite instability (MSI). This limitation of the work may have played an important role in the response of the cells to MCC knockdown, or SN38/Olaparib treatment. A previous study (reference 33 in this manuscript) demonstrated that manipulation of a number of cancer-associated factors altered the sensitivity of cells to irinotecan and rucaparib (another PARP inhibitor). Is it possible that the relevance of MCC methylation profiling or loss of MCC expression as a biomarker for sensitivity to irinotecan therapy is related more strongly to or critically influenced by other cancer-specific alterations (i.e. is it an independent predictor of responsiveness)?

Author Response

Response to Reviewer 1.

  1. In the Abstract (line 28), the authors state that SN38/irinotecan treatment increases DNA damage. Where are these data – they do not seem to be included in the figures or text (is it possible that some of the files have not uploaded?).

The reviewer is correct and we apologise for overlooking this. The increase of DNA damage with SN38 treatment was shown in our previous publication (Figure 4F; Currey et al. Mouse model of MCC deletion reveals novel pathways in colon inflammation and cancer. Cell Mol Gastroenterol Hepatol 2019; 7: 819-839). The reference to DNA damage has been removed from the abstract. This result has already been correctly referenced elsewhere in the paper.

  1. In Figure 1B, have the authors included data for ‘MCC deep deletion’, or are the values so low that they not visible in the bar graph?

The values for MCC deep deletion are indeed so low that they cannot be seen in the bar graph.

  1. The authors seem to have used a confluency assay to estimate cell proliferation in Figure 2A, but then in figures 2B and 2C where they are estimating IC50 values, they use Alamar Blue. Is there any reason for using 2 different assays? Does MCC knockdown alter cell size, cell spread on the culture surface or contact inhibition?

Figure 2A is only included as a general characterisation to show that MCC-knockdown increases cell confluency/proliferation. We have rearranged the sections in Materials and Methods to reflect this. MCC knockdown does indeed alter cell-cell adhesion properties, and we have published on this previously (Benthani et al. Oncogene 2018; 37(5): 663-672.)

  1. The resolution of images in Figure 2F is not suitable for publication. Images are out of focus and it is not possible to discern the nuclear or cytoplasmic localisation of PARP that the authors mention in the text. Could these images please be replaced with images that are in focus – a slightly higher magnification may also help.

Unfortunately we are not able to submit a higher magnification from this experiment. Instead we have included a larger image for some cells as additional panels and moved the whole enlarged figure into supplementary files (Figure S3).

  1. In Figure 3A, the authors show Western blot images for DNA-PKc and pDNA-PKc, but have only included the blot for pATR. Please add the matching blot for ATR as results are difficult to interpret without this.

The quality of the ATR blots was variable. Therefore, we show this result in the supplementary files only. We did not detect a clear change in the level of basal ATR protein expression between MCC-WT and MCC-KO cells. As for pATR, we have since been made aware that the S428 phosphosite should not be used as a readout of ATR activity (Blackford AN and Jackson SP 2017. Mol Cell 66:801). Therefore, we have omitted the pATR-S428 western blot from the manuscript.

  1. I am not sure that I understand how the authors are quantitating their Western blots. It is usual to normalise each band to its respective b-actin. However, to understand whether a treatment is inducing phosphorylation of a particular protein, the (b-actin normalised) phosphorylated protein level is divided by the (b-actin normalised) level of the corresponding total protein. Could the authors please clarify that this is what they have done as it is not clear from the figure legends or Methods section (Figure 3).

We always normalise each band to the matching loading control. We apologise that we had not done the ratio of b-actin normalised pDNA-PKc and b-actin normalised DNA-PKc. This has now been included in Figure 3A and the text has been corrected.

  1. I feel that the limitations of the study and the limitations of the experimental design have not been clearly enunciated and that these should be added to the manuscript. In particular, use of a single cell line is a major limitation of the study that needs to be clearly defined. While creation of biological replicates enables additional points to be added to graphs, it does not diminish the limitation of sole use of one extensively cultured cell line with its single genetic background. Reasons for the lack of verification of any results with additional cell lines should also be added. The amount of time (number of passages) that control and MCC-deleted sublines were cultured should also be added to the Methods section.

We have shown a similar effect with irinotecan/SN38 in two cell lines, HCT116 and MEF. We selected the HCT116 cell line because we wanted to test the impact of MCC deletion on the background of CIMP-positivity. MEF cells represent a non-cancer cell line. We agree that use of further cancer cell lines would strengthen the conclusions. We have added a comment about this limitation in the Conclusions section.

The number of passages (~20) has been added into Materials and Methods.

  1. The number of replicates used for experiments and the number of times that experiments were repeated are difficult to follow. For example, the authors state that they only used 3 mice per group, however I cannot find a statement detailing how many times experiments were repeated. In addition, I cannot find descriptions regarding how consistent results from the biological replicates were. Was this the reason why all biological replicates were not used in experiments?

The biological replicates of each cell line gave consistent results but we only used 4-6 biological replicates for most of the final images. For PARP protein expression in Figure 3A, statistical comparisons were done from three independent experiments with 3 biological replicates. A comment has been added in the Materials and Methods that the drug treatment experiments were performed at least three times.

  1. In this study, the authors have made several interesting observations regarding MCC downregulation and irinotecan/PARP inhibitor sensitivity. However, I feel that the conclusions of the authors may be overstated at this time. Firstly, I don’t feel that MCC methylation profiles have been evaluated sufficiently/specifically by the authors in relation to their potential use as a marker of CIMP. In addition, HCT116, the only colon cancer cell lines used in the study, carries a particular set of mutations and has microsatellite instability (MSI). This limitation of the work may have played an important role in the response of the cells to MCC knockdown, or SN38/Olaparib treatment. A previous study (reference 33 in this manuscript) demonstrated that manipulation of a number of cancer-associated factors altered the sensitivity of cells to irinotecan and rucaparib (another PARP inhibitor). Is it possible that the relevance of MCC methylation profiling or loss of MCC expression as a biomarker for sensitivity to irinotecan therapy is related more strongly to or critically influenced by other cancer-specific alterations (i.e. is it an independent predictor of responsiveness)?

We have removed the strong statement that MCC methylation is a marker of CIMP (Conclusions) because not all MCC-methylated tumours are classified as CIMP-H. Our previous published data on two independent patient cohorts (Oncogene 2007, 26, 4435-4441; Cancers 2021, 13, 3529) as well as the TCGA cohort (this study) show that CIMP-H cancers are almost always MCC-methylated but not the other way around.

As noted above we have now stated in the conclusions that further cancer cell lines need to be investigated. We have also added a statement in the discussion which highlights reference 33 and the finding that Irinotecan/PARPi sensitivity is known to be affected by several factors.

We thank the reviewers for their insights and valuable suggestions and hope that our revised manuscript is acceptable for publication in Cancers.

Reviewer 2 Report

The manuscript "MCC gene silencing is a CpG island methylator phenotype-associated factor that predisposes colon cancer cells to irinotecan and olaparib" indicates MCC as predictive biomarker for irinotecan response in colon cancer.

Overall the study is well organized and methods are extensively described. The conclusions are suggested by the results, but further studies are needed to support and to confirm them.

Major points:

  1. Figure quality must be improved
  2. Checking the Western blot file provided by the authors, in the first blot (pDNA-PKc) it is possible to see a non specific signal that is identical of that the authors indicated as pATR in the Figure 3A. This reviewer is not sure that the signals represent pATR.
  3. methods: line 41 pag2. the authors said: "Six clonal MCC-deleted HCT116 cell lines were pooled after their first few passages and then split into seven biological replicates". Why? Why were different cloned pooled together and then split?
  4. lines 35-36 pag.8 Figure 2 F. For this Reviewer it is difficult to understand how PARP has been localized into the cytoplasm in MCC-WT cells. Normally PARP is a nuclear protein
    (https://www.proteinatlas.org/ENSG00000143799-PARP1/subcellular).
    Is it this localization  dependent on irinotecan treatment?
  5. line 42 pag.8 Given that DNA-Pk is shown in Fig 1A, is it possible to add also total ATR and RAD51 (herein described as not shown data)?
  6. Figure 3F. What is the pvalue refered to?
  7. the authors detected and analyzed cleaved form of parp. Could they comment it describing the meaning?

Minor points:

  1. The first line of introduction needs an appropriate reference.
  2. line 44 pag4 spell out CIMP-L
  3. FiG.2 several panels reported pvalue but it is not specified the specific points of graphs between which the pvalue was calculated (IC50?, 200mm3?)

Author Response

Major points:

  1. Figure quality must be improved

The figures had lost resolution when they were embedded to the journal template. We have inserted better quality images in the manuscript.

  1. Checking the Western blot file provided by the authors, in the first blot (pDNA-PKc) it is possible to see a non specific signal that is identical of that the authors indicated as pATR in the Figure 3A. This reviewer is not sure that the signals represent pATR.

We thank the reviewer and apologise for our oversight. We repeated the ATR/pATR and DNA-PKc/pDNA-PKc blotting several times but the pATR image chosen for Figure 3A was poor quality. We are confident that the DNA-PKc/pDNA-PKc blots are correct, but the ATR/pATR antibodies have been variable and often produce non-specific bands, making quantification difficult. We have included an ATR blot from a different gel in the supplementary files. This still shows some cross-reactive bands but indicate no difference in the basal ATR expression between MCC-WT and MCC-KO cells. As for pATR, it has been pointed out that the S428 phosphosite should not be used as a readout of ATR activity (Blackford AN and Jackson SP 2017. Mol Cell 66:801). Therefore, we have omitted the mention of ATR activation from the main manuscript text.

  1. methods: line 41 pag2. the authors said: "Six clonal MCC-deleted HCT116 cell lines were pooled after their first few passages and then split into seven biological replicates". Why? Why were different cloned pooled together and then split?

We pooled the MCC-deleted cell lines because the control MCC-expressing lines were not clonal origin. Subsequently, we split both MCC-WT and MCC-KO HCT116 cells into parallel cell lines. Both MCC-WT and MCC-KO cells were always maintained as 6-7 separate lines and matching passages were used for experiments.

  1. lines 35-36 pag.8 Figure 2 F. For this Reviewer it is difficult to understand how PARP has been localized into the cytoplasm in MCC-WT cells. Normally PARP is a nuclear protein
    (https://www.proteinatlas.org/ENSG00000143799-PARP1/subcellular).
    Is it this localization  dependent on irinotecan treatment?

Indeed, nuclear PARP is more prominent than cytoplasmic PARP in normal cells. Cytoplasmic PARP1 has been shown to occur in several cancer cell lines, for example, pancreatic cancer cells (PMID: 30614530) and breast cancer cells (PMID: 24535158). We have moved this image into supplementary files because of the large size of the new modified image.

  1. line 42 pag.8 Given that DNA-Pk is shown in Fig 1A, is it possible to add also total ATR and RAD51 (herein described as not shown data)?

Total RAD51 data has now been added to Figure 3A and the ATR is shown in the supplementary files.

  1. Figure 3F. What is the p value refered to?

The P value shown in the graph refers to one-way repeated measures ANOVA for drug doses 0.5-4. The pair-wise comparisons are also significant for all drug doses, SN38 vs SN38+olaparib P=0.026; olaparib vs SN38+olaparib P=0.010.

  1. the authors detected and analyzed cleaved form of parp. Could they comment it describing the meaning?

The appearance of a PARP cleavage fragment is an indication that a proportion of the cells are undergoing apoptosis. This has been added in the figure legend.

Minor points:

  1. The first line of introduction needs an appropriate reference.

The first sentence has been corrected and a reference has been added.

  1. line 44 pag4 spell out CIMP-L

This has been corrected.

  1. FiG.2 several panels reported p value but it is not specified the specific points of graphs between which the p value was calculated (IC50?, 200mm3?)

In Figure 2B and C the P value refers to the comparison of all drug doses (paired t test), not only IC50. In Figure 2D also, the P value refers to all time points in the left panel and the last four time points in the right panel, as mentioned in the legend.

We thank the reviewers for their insights and valuable suggestions and hope that our revised manuscript is acceptable for publication in Cancers.